# Advancing Cancer Therapy with Copper/Disulfiram Nanomedicines and Drug Delivery Systems

**DOI:** 10.3390/pharmaceutics15061567

**Published:** 2023-05-23

**Authors:** Xuejia Kang, Sanika Jadhav, Manjusha Annaji, Chung-Hui Huang, Rajesh Amin, Jianzhong Shen, Charles R. Ashby, Amit K. Tiwari, R. Jayachandra Babu, Pengyu Chen

**Affiliations:** 1Department of Drug Discovery and Development, Harrison College of Pharmacy, Auburn University, Auburn, AL 36849, USA; 2Materials Research and Education Center, Materials Engineering, Department of Mechanical Engineering, Auburn University, Auburn, AL 36849, USA; 3Department of Pharmaceutical Sciences and Experimental Therapeutics, College of Pharmacy, University of Iowa, Iowa City, IA 52242, USA; 4Department of Pharmaceutical Sciences, College of Pharmacy, St. John’s University, Queens, NY 11431, USA; 5Department of Pharmacology and Experimental Therapeutics, College of Pharmacy and Pharmaceutical Sciences, University of Toledo, Toledo, OH 43614, USA

**Keywords:** disulfiram/copper, cancer, nanomedicines, drug delivery systems, cuproptosis, immunomodulatory effects

## Abstract

Disulfiram (DSF) is a thiocarbamate based drug that has been approved for treating alcoholism for over 60 years. Preclinical studies have shown that DSF has anticancer efficacy, and its supplementation with copper (CuII) significantly potentiates the efficacy of DSF. However, the results of clinical trials have not yielded promising results. The elucidation of the anticancer mechanisms of DSF/Cu (II) will be beneficial in repurposing DSF as a new treatment for certain types of cancer. DSF’s anticancer mechanism is primarily due to its generating reactive oxygen species, inhibiting aldehyde dehydrogenase (ALDH) activity inhibition, and decreasing the levels of transcriptional proteins. DSF also shows inhibitory effects in cancer cell proliferation, the self-renewal of cancer stem cells (CSCs), angiogenesis, drug resistance, and suppresses cancer cell metastasis. This review also discusses current drug delivery strategies for DSF alone diethyldithocarbamate (DDC), Cu (II) and DSF/Cu (II), and the efficacious component Diethyldithiocarbamate–copper complex (CuET).

## 1. Introduction

There has been tremendous progress in the fields of drug discovery, tumor biology, nanomedicine, and targeted drug delivery for improving treatment and patient care, yet cancer remains one of the leading causes of death. The process of discovering new chemical entities and their development into anticancer drugs can be very time consuming and expensive. Drug repurposing via new drug delivery systems is a relatively less time-consuming, simple, and cost-effective strategy in treating many diseases [1,2]. With the advancement of computational and rapid screening technologies, new drug candidates are now available for cancer treatment. However, the efficient delivery of these drugs to cancer tumors still requires novel drug delivery devices and methods [3,4].

Copper stimulates the proliferation and migration of endothelial cells and is required for the secretion of several angiogenic factors by tumor cells [5]. However, copper chelation has been reported to produce a decrease in the secretion of many of these factors [5,6]. Recently, Based on clinical data indicating that elevated serum copper levels associated with many cancer tumors, many copper chelators are being developed and tested in clinical trials in recent years [7]. Although it remains to be determined the mechanism which copper chelation suppresses the growth of endothelial cells and hinders the secretion of angiogenic factors by tumors is not entirely clear, it is has been hypothesized to result from effects on copper-dependent enzymes, transporters, and chaperones [8].

Recently, a novel cell death pathway, distinct from apoptosis, necroptosis, pyroptosis, and ferroptosis, induced by copper, has been identified, and designated as cuproptisis [9]. Cuproptosis is significantly correlated to cellular metabolism and is frequently observed in certain types of cancer that have high levels of aerobic respiration, including melanoma, breast cancer, leukemia, and drug-resistant cancers [10]. Copper ionophores have played a crucial role in the identification of cuproptosis and have been considered as possible anticancer treatments [11,12]. Diethyldithiocarbamate (DDC), an active metabolite of the drug, disulfiram (DSF), has been reported to be a copper ionophore that has in vitro and in vivo efficacy as an anticancer compound. [11].

Disulfiram (DSF,) is a thiocarbamate derivative that has been used to treat alcoholism since 1951 [13]. DSF inhibits the enzyme aldehyde dehydrogenase 1 (ALDH1), which significantly inhibits the biotransformation of ethanol to ethanol at the acetaldehyde stage [14], thereby increasing the levels of ALDH1, which produces various adverse effects following the ingestion of ethanol, decreases or discourage ethanol intake [15]. Interestingly, ALDH1 is present found in cancer cells with stem cell properties, and it catalyzes the oxidation of intracellular aldehydes and produces multidrug resistance [16]. Subsequently, clinical data suggested that DSF had anticancer efficacy [17]. In addition, DSF was shown to regulate the balance of reactive oxygen species (ROS) and glutathione (GSH) [18], inhibit the activity of the ubiquitin proteasome system (UPS) [19], and regulate intracellular signaling [20,21] and the activity of other enzymes, which could play a role in mediating its anticancer efficacy [22,23,24]. Indeed, these mechanisms could induce cancer cell death, decrease the stemness of cancer cells [25], decrease angiogenesis, and overcome drug resistance [26,27].

As mentioned earlier, trace metals are essential for the survival of cancer cells [28,29]. Compared with normal cells, many tumor cells have a 2–3-fold higher concentration of copper [30]. DSF is a potent chelator of copper, and DSF can bio-transform the pro-angiogenic activity of copper to a specific compound that induces cancer cell death [31]. Although DSF has shown promise in both laboratory and animal studies, certain clinical trials with cancer patients have been unsuccessful due to treatment requiring high doses and the expectation partially owing to the rapid biodegradation of the DSF [32,33]. Cu (II) chelation of the primary DSF metabolite, diethyldithiocarbamate (DDC), is crucial for inducing the death of tumor cells [34,35]. The copper-dependent anticancer efficacy of DSF has resulted in an increase in research related to DSF/Cu (II) [36,37]. DSF/Cu (II) has been reported to be efficacious in a variety of cancers, including liver cancer [38], ovarian cancer [39], prostate cancer [25], lung cancer [40], glioblastoma (GBM) [41], and breast cancer [42] (Figure 1). The anticancer efficacy of DSF is most likely due to the bio-transformation of DSF to diethyldithiocarbamate (DDTC), which forms an intracellular copper–DDTC complex (Cu(DDTC)_2_), which has been shown to be the most efficacious anticancer compound [43]. DSF can induce ferroptosis and cuproptosis in different types of cancer cells [44]. A summary of various clinical trials assessing the efficacy of DSF-based cancer is shown in Table 1. Some clinical trials have investigated the use of DSF for treating solid tumors, with some reporting improved progression-free survival and overall survival, compared the control groups. Adverse effects were generally mild and resolved following a decrease in the dose of DSF. Two single-arm trials in glioblastoma patients showed positive effects, while a randomized controlled trial in NSCLC patients also reported an increase in patient survival [45,46,47]. However, the response to DSF varied among patients, and further in vitro and animal studies are needed to explore the optimal concentration and sensitivity type. Overall, DSF appears safe and effective in prolonging survival in cancer patients [48].

## 2. Anticancer Mechanisms of DSF/Cu (II)

It has been revealed that the anti-cancer mechanism of DSF/Cu (II) may be mediated by the regulation of reactive oxygen species (ROS), enzyme activity regulation, induction of DNA damage, proteasome inhibition, and transcription factors [24] (Figure 2). Additionally, DSF/Cu (II) also exhibits immunomodulatory effects on tumor microenvironment (TME).

### 2.1. Disulfiram/Cu with ROS

The effects of ROS on cancer cells has been reported to becontext-dependent [49]. For example, in the initial stage of cancer development, ROS levels are100-fold higher in cancer cells compare to normal cells [22,50]. The increased levels of ROS activates signaling transduction pathways in cancer cells, regulating the activities of redox-sensitive transcription factors [22,50], and facilitating cancer cell survival and proliferation [51,52]. However, in contrast, the excessive ROS that is more than the tolerated limit of cancer can kill the cancer via cell cycle arrest and apoptosis, so the increase in the basal level of ROS in cancer cells will benefit the use of ROS-inducing anti-cancer agents [51,53].

In vitro, 1 μM of DSF was reported to increase the cytotoxic efficacy of cisplatin by increasing the accumulation of ROS [54]. The chelation of DSF with Cu (II)produces a further increase in the production of ROS [42,55,56]. The complex of DSF and Cu (II)has been reported to inhibit the enzyme superoxide dismutase 1 (SOD1), one of the major enzymesthat mitigates oxidative damage in melanoma cells [57]. The inhibition of SOD1 increases the formation of superoxide, thereby increasing oxidative stress [57]. Lipid ROS levels also increased in the presence of Zn (II) and DSF [57,58]. The increased ROS levels were shown to be essential for DSF-induced apoptosis in melanoma cells [42]. The inhibition of Glutathione reductase (GSR) inhibition by DSF disrupts glutathione GSH redox cycling, producing an accumulation of oxidized glutathione (GSSG) and a lower GSH/GSSG ratio, producing an increase in ROS level [59,60].

### 2.2. Enzyme Inhibition and DNA Damage

As stated earlier in this article, the presence of ALDH is an indicator of cancer stem cell (CSC) activity, which is significantly correlated with tumor progression, metastasis, and drug resistance [61]. Furthermore, the inhibition of the Hedgehog pathway and the proteins Nanog and Oct-4 (the cell stemness transcription factors) result from DSF/Cu (II) decreasing the number of CSCs, thereby inhibiting the proliferation of myeloma cells [62]. DSF inhibits the metastasis of osteosarcoma cells via irreversibly inhibiting ALDH [63,64,65]. DSF suppresses the expression of DNA topoisomerases, DNA methyltransferase, DNA polymerase, and ribonucleotide reductase [66,67]. O6-methylguanine-DNA methyltransferase (MGMT), a unique antimutagenic DNA repair protein [68,69], has been reported to be overexpressed in brain tumors [70]. Disulfiram is a direct inhibitor of MGMT, which greatly increases cancer cell death [71]. Interestingly, DSF also inactivate the enzyme phosphoglycerate dehydrogenase (PHGDH) by covalently reacting with cysteine residues that mediate DNA repair [71,72,73]. DNA damage may also be facilitated by therapeutics- induced ROS generation, which involves the opening of pores in the mitochondria [58,61]. The BRCA1 and BRCA genes are essential for the repair of double-stranded DNA breaks and homologous DNA recombination [74,75]. Individuals who have BRCA mutations are more likely to develop breast cancer comparedto people who undergo no BRCA mutations [74,75], in vitro experiments indicate that the cell viability of BRCA mutant cell lines can also be significantly decreased by DSF/Cu (II) [76]. 

### 2.3. The Effect of DSF/Cu (II) on the Activity of the Proteasome System 

DSF also exerts anticancer efficacy by the inhibiting of the ubiquitin proteasome system (UPS). The UPS is crucial for the balance of protein metabolism and the normal function of cells [77]. Studies indicate that the proliferation of cancer cells is altered by proteasome inhibition [78]. The 26S proteasome consists of a catalytic 20S core and a 19S regulator, which biodegrades ubiquitinated proteins [79,80]. The active sulfhydryl groups in DSF chelate biological metals such as copper, which subsequently inhibit the 26S proteasome [81]. Importantly, the inhibition of the UPS by DSF/Cu (II), specifically occurs in breast cancer and prostate cancer cells [82] but not in normal cells [81,82]. Recently, a new mechanism of the DDC/Cu (II) complex CuET inhibiting the UPS has been reported to be associated with the proteins p97 and NPL4 [34]. NPL4 contains a Zn finger domain with two zinc fingers, and bivalent metal ions, such as copper, can bind to Zn fingers and cause the aggregation of NPL4 [83]. Furthermore, VCP/p97 interacts with the aggregated NPL4, causing an impairment of the p97-NPL4-UFD1 pathway [34]. All these mechanism consequently block substrate biodegradation, and induce cancer cell death CuETs have been reported to have greater antitumor efficacy than DSF [58,84], as CuET binds NLP4 with a higher affinity than DSF and induces NLP4 aggregation [34]. In addition, DSFalone is a proteasome inhibitor, which means it inhibits the activity of proteasomes, cellular structures responsible for protein degradation, interestingly, DSFalso converts carcinogenic cadmium to a proteasome inhibitor that possesses pro-apoptotic activity in human cancer cells [85]. NF-κB activity has been reported to be associated with the inhibitory activity of DSF on the proteasome [86]. DSF decreases NF-κB activity by blocking proteasome activity [87]. This occurs through the proteasomal cleavage of the protein inhibitor-κB (IκB) [88], which releases the heterodimer p50/p65 from the inhibitory complex, allowing for the translocation of NF-kB to the nucleus, where the regulation of the transcription of certain genes, notably those that mediate inflammation and cell survival happens [87,89]. In vitro, 10 μM of DSF/Cu (II) completely inhibited the chymotrypsin (CT)-like proteasomal activity in glioma stem cells from patients with multiple glioblastoma multiforme [90].

### 2.4. The Effect on Transcription Factors

DSF form mixed sulfides with sulfhydryl-containing transcription factors (TFs) and prevent the proliferation of melanoma cells [35,91]. NF-kB is a typical TF that mediates the development and progression of cancers [92]. Due to the cysteines in the DNA binding region of NF-kB, DSF affects cancer cells by binding in the DNA binding region of NF-kB and blocking its pro-cancer effects [35,93,94]. The presence of metal ions, such as Cu (II) and Zn (II), increases the formation of mixed sulfides [95]. NF-kB induces the expression of antiapoptotic genes, as indicated by the aberrant activation of NF-kB in many human malignancies [92]. Thus, the inhibition of NF-kB via DSF could increase cancer cell apoptosis [96]. DSF/Cu (II) is efficacious in selectively eradicating leukemia stem cells by simultaneously activating the apoptosis-related cJun N-terminal kinase (JNK) [96].

### 2.5. The Immunomodulatory Effects on the Tumor Microenvironment (TME)

The tumor microenvironment is a promising target for cancer therapy [97]. M2 tumor-associated macrophages (TAMs) are an important tumor-promoting component in the TME [98]. In vitro, Cu/DSF inhibits M2 TAM recruitment and to reprogram M2 TAMs by regulating the expression of certain chemokines and cytokines [99]. FROUNT (also be known as NUP85), a cytoplasmic protein, interacts with the chemokine receptors CCR2 and CCR5 in macrophages, facilitating the recruitment of TAMs [100]. Furthermore, FROUNT activates the PI3K-Rac-lamellipodium cascade, promoting cancer proliferation [100]. DSF has been reported to inhibit the interaction of FROUNT with CCR2 and CCR5, which decreases tumor associated macrophage (M2) accumulation in the TME, thereby decreasing tumor-promoting properties of tumor associated macrophage [100]. The co-delivery of the compound honokiol and a DSF/Cu (II) complex reprograms M2 TAMs to anti-tumor M1 macrophages and regulates mTOR [41]. In addition, DSF/Cu (II) and CuET induce immunogenic cell death (ICD), as indicated by a significant increase in the expression of damage-associated molecular pattern molecules, such as calreticulin, ATP, and high mobility group box 1 protein, in 4T1 breast cancer [101], colorectal cancer [102,103], and hepatocellular carcinoma [104,105]. The dying cancer cells become an in-situ vaccine that elicit significant immune memory by activating dendritic cells and the CD8 T cell immune response [104,105]. 

## 3. The Effect of DSF/Cu (II) on Cancer 

In this section, we summarize the effects of DSF/Cu (II) on cancer-associated activities such as unlimited cancer proliferation, the self-renewal activity of cancer stem cells, cancer angiogenesis, and drug resistance (Figure 3).

### 3.1. DSF/Cu (II) on the Inhibition of Cancer Proliferation

DSF/Cu (II) induces cell death in different cancer cell lines via different mechanisms. In human breast cancer cells, DSF/Cu (II) causes cancer cell apoptosis via increasing Bcl-2 Associated X-protein (Bax, a pro-apoptotic protein) [50]. DSF, at a concentration of 25–50 ng/mL, produces a 4–6-fold increase in apoptosis, and co-incubation with the ROS inhibitory compound N-acetyl-cysteine (NAC) reverses DSF-induced apoptosis, suggesting that DSF-induced apoptosis is associated with an increase in ROS levels [60]. DSF induces the disruption of the mitochondrial membrane potential and cause apoptosis in human melanoma cell lines [60]. Xi et al. reported that DSF/Cu (II) produced significant cytotoxicity and caspase-dependent apoptosis in NSCLC cells [106]. Additionally, DSF produced autophagy-dependent apoptosis [103]. DSF/Cu (II) induced the apoptosis of erbB2-positive breast cancer cells by inhibiting AKT, cyclin D1, and NFκB signaling [107]. In malignant pleural mesothelioma (MPM) cells, DSF/Cu (II) produced apoptosis by activating the proapoptotic stress-activated protein kinases (SAPKs) p38 and JNK1/2, caspase-3. Furthermore, DSF/Cu (II) increased the expression of the apoptosis transducer, cell division cycle, and apoptosis regulator 1 (CARP-1/CCAR1) and sulfatase 1 (SULF1) [108]. The activation of NF-κB mediates cancer proliferation [109] and the DSF/Cu (II) inhibits the activity of NF-κB, thus inhibiting hepatocellular carcinoma (HCC) growth [110].

### 3.2. DSF/Cu (II) Efficacy in Cancer Stem Cells (CSCs)

CSCs have been reported to mediates cancer angiogenesis and drug resistance [111], so we firstly discuss the functions of DSF/Cu (II) on CSCs, then, in the following two paragraphs, we will discuss the roles of DSF/Cu (II) on angiogenesis and drug resistance. Because of promotions of CSCs on activities including the activation of ABC transporters, such as P-gp [112], and the increase in DNA repair mechanisms [113], CSCs are resistant to conventional anticancer drugs [37]. Furthermore, hypoxia also induces resistance of CSCs to chemotherapy. Numerous studies indicate that the presence of CSCs in patients correlates with poor prognosis [37]. The roles of DSF/Cu (II) on CSCs can be attributed to its activity on the ALDH1 enzyme. The ALDH1 protein family (ALDH1A1, ALDH1A2, and ALDH1A3) enhance the self-renewal, survival, and proliferation of CSCs [114]. ALDH+ CSC phenotypes have a high tumorigenic capacity [115,116]. DSF/Cu (II) significantly decreases MDA-MB-231 breast cancer cell proliferation by decreasing the ALDH+ CSC population [117]. In hepatocellular carcinoma, DSF decreases CSCs by inhibiting the p38 mitogen-activated protein kinase (MAPK) pathway [118]. DSF also inhibits CSCs in ovarian, pancreatic, pulmonary, and hematological cancers [90,119,120].

### 3.3. DSF/Cu (II) Effects on the Inhibition of Cancer Angiogenesis

Vascular endothelial growth factor (VEGF) is a critical mediator of angiogenesis in cancer cells, and high VEGF levels are positively correlated with poor prognosis [121]. 4HNE, a lipid peroxidation product [122], is involved in angiogenesis [123,124]; and the catabolism of 4HNE is dependent on the cellular levels of glutathione-S-transferases, alcohol dehydrogenases, and ALDHs [125]. The inhibition of ALDH2 by DSF/Cu (II) significantly decreases angiogenesis by inhibiting the hypoxia-inducible factor-1α (HIF-1α)/VEGF signaling cascade [126]. The inhibition of SOD-1 by DSF/Cu (II) induces endothelial cell growth arrest and apoptosis and, thus, exhibits anti-angiogenesis efficacy [127]. DSF/Cu (II) also inhibits tumor angiogenesis by inhibiting the activity of matrix metalloproteinases [128,129]. Copper increases the anti-angiogenic efficacy of DSF via the EGFR/Src/VEGF pathway in gliomas [130]. CSCs were shown to modulate angiogenesis via CSC-secreted VEGF [131]. Moreover, CSCs overexpress CXCR4, whose SDF-1/CXCL12 ligand induces VEGF production via activation of the P13K/AKT signaling pathway [132]. Other CSC-associated factors such as SDF-1/CXCL12 also play roles in the formation of the new blood vessels [133,134]. Thus, the inhibition of DSF/Cu (II) in CSCs decrease angiogenesis.

### 3.4. DSF/Cu (II) Reverses Drug Resistance

DSF/Cu (II) overcomes drug resistance via targeting the proteasome, epithelial–mesenchymal transition (EMT), P-gp, CSC activity [135,136,137]. By targeting the proteasome, DSF significantly increases the sensitivity of TMZ-resistant brain tumor-initiating cell (BTIC) variants (BT73R and BT206R) to temozolomide (TMZ) [138]. Numerous studies have shown that EMT plays a role in mediating the resistance of cancer cells to certain anticancer drugs, such as paclitaxel in prostate (DU145-TXR) and lung cancer (A549-TXR) [139,140,141]. By downregulating associated proteins such as Vimentin, DSF/Cu (II) inhibits the EMT, which consequently overcomes the paclitaxel resistance of prostate and lung cancer [141]. In addition, DSF/Cu (II) decreases the effects of EMT in breast cancer cells via the regulation of protein kinase (ERK)/NF-κB/Snail pathway [142]. Cancer cells expressing high levels of P-gp exhibit resistance to conventional chemotherapy drugs like doxorubicin and paclitaxel, a phenomenon referred to as multidrug resistance (MDR) [143,144]. However, the metabolite and active anti-cancer compound CuET is not a substrate of P-gp, and thus it is still retained inside of drug resistant cancer cells and increase the likelihood of the drug-resistant cancer cells death [141]. DSF/Cu (II) produce efficacy in osimertinib-resistant NSCLC cells by activating macrophage-mediated innate immunity [40]. It was discovered that the high levels of ABC protein provide the protective mechanism for CSCs to chemotherapeutics [145]. The inhibitory effects of DSF/Cu (II) on the CSCs benefit the overcoming of drug resistance [39]. Another important finding is that Cu(DDC)2 NP is also does not inhibit P-gp activity or expression, thus avoiding the side effects associated with P-gp inhibitors [141].

**Figure 3 pharmaceutics-15-01567-f003:**
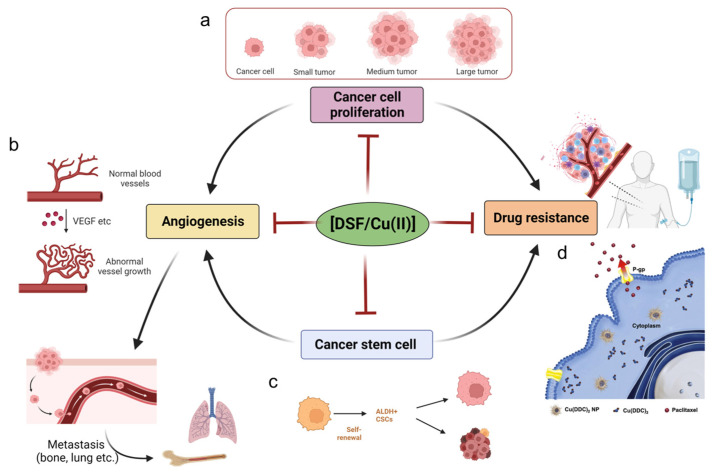
Cancer cells have unlimited proliferative capacity, and DSF/Cu (II) has been shown anti-proliferation effects towards cancer cells. (**a**) DSF/Cu (II) inhibits cancer cell proliferation, preventing the transformation of small cancer lesions to large tumors. Created with BioRender.com accessed on 8 May 2023 (**b**). Angiogenesis is an essential step for cancer metastasis; the VEGF in the cancer microenvironments contributes the massive and abnormal vessels in cancer lesions, and DSF/Cu (II) inhibits the angiogenesis behavior and prevents cancer metastasis to lung and bone, etc., sites. Created with BioRender.com. accessed on 8 May 2023 (**c**) Cancer stem cells aggravate the angiogenesis and drug resistance of cancer; DSF/Cu (II) inhibits cancer stem cells and thus shown potency in anti-angiogenesis and anti-drug resistance. Created with BioRender.com accessed on 8 May 2023 (**d**). Drug resistance is a typical phenomenon during the treatment of cancer; novel copper diethyldithiocarbamate nanoparticles can effectively overcome drug-resistant cancers, owing to being non-binding to P-gp and being maintained in the cancer cells. Copyright 2023, Elsevier [141].

## 4. DSF-Based Therapies for the Treatment of Cancer

Although data from in vitro studies suggested that DSF could be an efficacious anticancer treatment, clinical trials with oral DSF plus Cu (II) in cancer patients have been equivocal [146,147]. The underlying reasons for the poor clinical results include: (1) the instability of DSF in the gastrointestinal environment; (2) the rapid biodegradation of DSF via first-pass metabolism; and (3) low final Cu (II) concentration at the tumor sites [148]. The effective delivery of DSF and Cu (II) to target sites is crucial for maximizing the anticancer efficacy of DSF/Cu (II) and overcoming limitations such as poor solubility, stability, and bioavailability. Recent progress in nanotechnology has facilitated the targeted delivery of DSF, and various types of drug delivery systems based on different nanoparticles have been developed. For instance, polymeric nanoparticles, nanogels, polymer–drug conjugates, liposomes, and dendrimers have been explored as effective carriers for DSF. For instance, DSF-loaded vitamin E-TPGS-modified PEGylated nanostructured lipid carriers have gained significant attention due to their biodegradable and biocompatible properties [149]. Nanogels are composed of nanosized particles that can entrap and release drugs in response to different stimuli, providing a promising strategy for the targeted delivery of DSF [150]. The use of these nanoparticle-based formulations can increase the accumulation of DSF at the target site, thereby reducing the toxic effects on healthy tissues and improving the therapeutic index. Overall, nanotechnology-based strategies have shown promising results in enhancing the anticancer efficacy of DSF and can potentially overcome the limitations associated with conventional DSF-based therapies. Figure 4 explains and summarizes this with some reported examples. Representative examples of the above delivery system are elaborated in the following paragraphs, and additional important investigations are listed in Table 2. 

### 4.1. DSF Drug Delivery Systems and DDC Prodrug

To facilitate the use of DSF in breast cancer, Fasehee et al. designed novel poly(lactide-co-glycolide) (PLGA) NPs loaded with DSF, where DSF was released in a sustained manner [187]. In this study, the effect of 0.01 mg/mL DSF-PLGA NPs were evaluated in breast MCF-7 cancer cells, compared to free disulfiram, DSF-PLGA NPs showed more excellent anti-proliferation effects [187]. Cell-penetrating peptides (CPPs) are widely used for the increase in cargo’s cellular uptake [188]. To address non-specific internalization in normal cells, pH-sensitive lipid cell-penetrating peptide nanocapsules (DSF-S-LNCs) were sythesized. With the protective shielding materials (PEG-PGA) modification, the transactivator of transcription (TAT) peptides and drug were preferentially exposed in the acidic tumor microenvironment and exerted anti-HepG 2(Cu-enriched cancer cells) effects at DSF (3 μM) (Figure 5a) [160]. The higher concentrations of ROS (up to 100 μM) in tumors benefit the application of ROS-activatable prodrug. A prodrug of disulfiram (DDC) called DQ has been synthesized [101]. Though DQ will cause cell death with the activation the ROS basal level on cancer cells is still not sufficient for the activation of prodrug. Thus, extra ROS triggered by other therapies is needed [189]. One approach to achieving this is by utilizing copper sulfide (CuS), a photosensitizer of photodynamic therapy (PDT) and photothermal therapy (PTT) and can generate the needed ROS [189]. Hence, we designed a DQ micelle and, meanwhile, used the CuS as the cooper source to generate the final product, Cu (DDC)_2_ for breast cancer inhibition; in this work, the DQ generated massive DDC with the fuel of ROS induced by the photodynamic effects with CuS. In addition, other products’ quinone methide (QM), released by DQ, consume GSH and, thus, further increase ROS [101,167], The Fenton-like reaction of Cu (II) also contributes ROS. This positive feedback causes the anti-tumor effects of ROS and further increases the release of DQ [101,167], with the additive effects. An excellent in vitro anti-cancer ability was achieved in doses of DQ at 2 μM (Figure 5b). It has been reported that cancer cells overexpress the enzyme γ-glutamyl transferase (GGT) in cancer cells [190]. Recently, a GGT-sensitive DDC prodrug was developed; after being specifically activated by GGT overexpressed by cancer cells, the metabolite of DSF, dithiocarbamate, selectively eliminates GGT-expressing cancer cells [168]. The in vitro anti-tumor efficacy of prodrug is very distinct, ranging from 800 nM in prostate 22 Rv1 cancer lines to over 15 µM in normal prostate PWR-1E cells. Cellular GGT activity was the reason for the obvious difference [168].

### 4.2. Drug Delivery Systems for Cu (II) and DSF/Cu (II)

In clinical trials, DSF is provided orally, but Cu (II) is critical for the efficacy of DSF [191]. It is critical that Cu (II) has a high affinity for its carrier to avoid an insufficient amount of Cu (II) in tumor sites. Liposomes can deliver the hydrophilic and hydrophobic drug [151]. In one study, copper oleate was synthesized and formulated as liposomes. This copper oleate liposome (Cu (OI)2-L) was obtained via alcohol injection and achieved 85% drug loading efficiency. Cu (OI)2-L exhibited a slow-release profile, indicated by >70% retained drug over 8 h incubation. Prolonged circulation time was confirmed in pharmacokinetic studies. Cu (OI)2-L plus DSF nanoparticles showed excellent antitumor efficacy in bearing a hepatoma mice xenograft model [180].

A new technique for breast cancer therapy has been developed using Cu (II)DSF in situ chelation. Cu (II)and DSF are encapsulated separately into PEGylated hollow mesoporous silica nanoparticles (HMSNs). The DSF@PEG/Cu-HMSNs nano system enhances DSF-based chemotherapy without systemic toxicity. It utilizes a tumor-specific pathway involving two mechanisms: pathway I to II and pathway I to III. In pathway I to II, DSF acts as a chelating agent, selectively binding with Cu (II) ions within the tumor site. In pathway I to III, Cu+ ions initiate a Fenton-like reaction with hydrogen peroxide (H_2_O_2_) in the tumor microenvironment, generating reactive oxygen species (ROS) for localized cytotoxicity. This combined approach achieves enhanced antitumor efficacy by targeting cancer cells while minimizing harm to healthy tissues (Figure 6a) [174]. Albumin is an FDA-approved material, and BSA was found to be a good carrier for DSF/Cu (II). The regorafenib (Rego) and DSF/Cu co-encapsulation showed potential in reversing multidrug resistance of HCT8/ADR cells. In this work, it was shown that paclitaxel (PTX) showed a slight cytotoxic effect even with 2 µM. However, the potent inhibitory effects were observed in combination therapy (DSF/Cu and Rego), with an IC50 of DSF/Cu about 0.47 µM. Furthermore, modifying the ligand so that it binds to the cancer cell receptor, MD206, decreased the IC50 of DSF/Cu (II) to0.3 µM. In addition, the CD206 was also highly expressed in the tumor-promoting macrophages, so the targeted nanoparticle interacted with 2 sites to produce their anticancer efficacy (“two birds one stone”). (Figure 6b).

### 4.3. Drug Delivery Systems for CuET

Since CuET was shown to be a pharmacologically active molecule of DSF/Cu (II) [34], research was initiated to deliver CuET to target sites. Recently, a nanoliposomal (LP) CuET formulation was synthesized using ethanol injection as a facile, one-step method that is suitable for large-scale manufacturing [193]. IC50 of LP-CuET on YUMM 1.7 mouse melanoma cells was 91.39 ± 4.98 nM [193]. In 2017, a CuET-loaded liposome was synthesized, and. In 2017, a CuET-loaded liposome was synthesized, the intravenous administration of this formulation produced a 45% decrease of tumor burden in MV-4-11 human biphenotypic B myelomonocytic leukemia xenograft mice model; [182,186]. Although the above-mentioned strategies could be used for CuET delivery, the efficiency of drug loading is not optimal. To overcome the drug loading problem, a stabilized metal ion ligand complex (SMILE) technology was proposed by Chen et al., in which a high CuET drug loading can be achieved [25]. Compared with the traditional thin-film dispersion method, this approach increased drug concentration by over 200-fold. Additionally, the IC50 of Cu (DDC)_2_ NPs on paclitaxel-resistant prostate cancer cells (DU145-TXR) was only around 100 nM [25]. Subsequently, a scale-up and commercialization 3D-printed microfluidic device was developed to scale the production of CuET [185]. This device could achieve a flow rate of 2 mL/min. Since it is a continuous production device, it can produce around 5000 mg of SMILE nanoparticle drug formulation per day [185] (Figure 7). Delivery strategies, such as the SMILE method, could also be used for the delivery of other metal-based chelators.

### 4.4. The Advantages and Disadvantages of Different Drug Delivery Systems

As we have discussed, additional Cu (II) supplementation is needed for DSF alone and DDC prodrug delivery systems. The disadvantages or problems associated with DSF and DDC prodrugs are: (1) the potential mismatch of the pharmacokinetics of DSF or prodrug and Cu (II) in systematic application; (2) difficulties in estimating accurate amounts of additional copper for the DSF and DDC prodrug chelation process in vivo. The major issues for the separate delivery of DSF and Cu are the pharmacokinetic and pharmacodynamic profiles because of the non-simultaneous delivery of two components to the tumor sites. The co-delivery of DSF/Cu (II) is a better strategy due to the synchronous delivery of DSF and Cu (II). However, the loading of drugs requires further improvement. Direct CuET delivery may represent a novel approach that decreases the dose needed and increases the therapeutic index, whereas the unknown effects of the direct use of CuET still need to be addressed (Figure 8).

## 5. Summary and Future Directions

Drug repurposing is a relatively cost-effective and promising strategy for the development of anti-cancer drugs. The in vitro, and preclinical evaluation of investigations of DSF-based delivery systems on cancer applications have dramatically increased in recent years. This review introduces the current targets and formulations of Cu and DSF and its metabolites as anticancer treatments. The effects of DSF/Cu (II) by affecting ROS production, cellular enzymes (e.g., ALDH), transcription factors (e.g., NF-κB), and proteasome activity (e.g., P97/NPL4) reveal the powerful anticancer activity of DSF/Cu (II). In addition, DSF/Cu (II) immunomodulates innate immune response and adaptive immune response; it can be used as a reagent to 1) reprogram macrophages and 2) as a therapeutic adjuvant for immunotherapy. 

Although the existing strategies have significantly contributed to the progress of DSF for cancer, some issues still need to be addressed. It is first necessary to determine the precise dose for in vivo studies as high doses of DSF cause side effects such as neuropathic pain. Furthermore, the side effects of CuET remain unknown due to the paucity of studies conducted with CuEt.

In addition, from the delivery perspective, the optimization of physicochemical properties of NPs is critical for decreasing the incidence of side effects and achieving targeted tumor treatment. The “3R” delivery principle is significant for formulation design and optimization. Briefly, 3R defined as effective delivery of multiple drugs to the right place, with the right dose, at the right time. For the right place, the targeting modification or in situ formation of CuET could be helpful in minimizing the amount of off-target drug release. For the right dose, the major concern is the accurate estimation of in vivo usage, but because of the heterogenous properties of different cancer types, the exact amount of drug is difficult to be determined. To ensure accurate dose delivery bya sperate delivery system, analytical methods for determining Cu (II) concentrations at the local regions would be useful. A device that releases a drug at the right time in response to a remote trigger could allow for on-demand drug release.

Although it has been more than half a century since the anti-cancer efficacy of DSF has been discovered, the clinical needs associated with DSF-based treatments have not yet been met. Technological advances, such as organ-on-a-chip or patient-derived organoids may provide an alternative to in vivo or animal studies using in vitro organoids. Finally new tumor-targeting modifications, such as antibodies, nanobodies, aptamers, peptides could increase the delivery of drug to tumors, thus increasing the efficacy and decreasing the incidence and severity of adverse effect.

Cancer chemotherapy-based (pre)clinical trials face significant challenges, including the need to address the reproducibility of research findings. In vitro assays often produce promising results, but in vivo results are frequently disappointing due to the variability of DDC levels in individuals and the heterogeneity of cancer cell lines. Advances in technology, such as the use of organs on a chip, can help to address these challenges by increasing the accuracy of in vitro biological systems. Additionally, there is a need for more rigorous standards and guidelines for conducting (pre)clinical research. This will help ensure that (pre)clinical research can be translated into effective therapies and interventions that benefit patients while minimizing potential risks and ethical concerns.

## Figures and Tables

**Figure 1 pharmaceutics-15-01567-f001:**
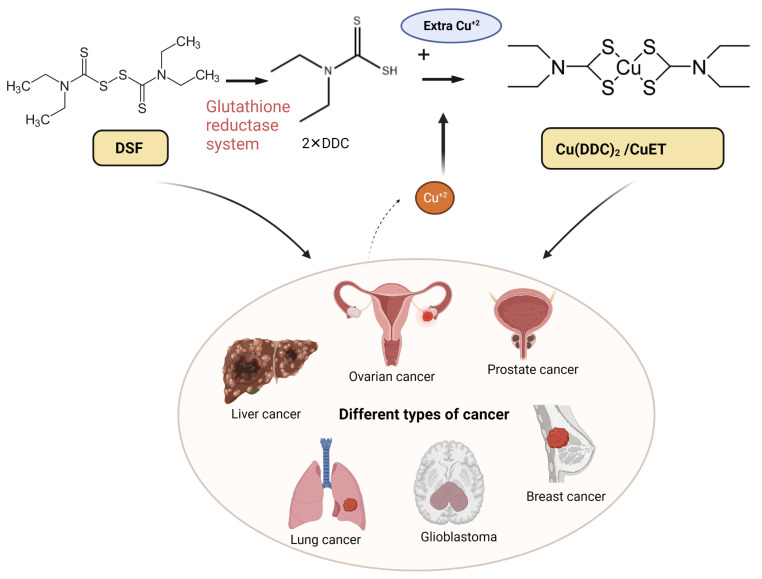
The chelation mechanism of DSF and Cu (II) and the use of DSF-based therapy for different types of cancer. DSF metabolizes to diethyldithiocarbamate (DDC or ET) via the glutathione reductase system; the active anti-cancer ingredient DDC further chelates with Cu (II) and forms Cu(DDC)_2_ (aka CuET), which has anti-cancer efficacy. The high dose of DSF alone and low dose of DSF/Cu (II) are effective in various cancers including liver, ovarian, prostate, breast, lung cancer, and glioblastoma (GBM). Created with BioRender.com. accessed on 19 April 2023.

**Figure 2 pharmaceutics-15-01567-f002:**
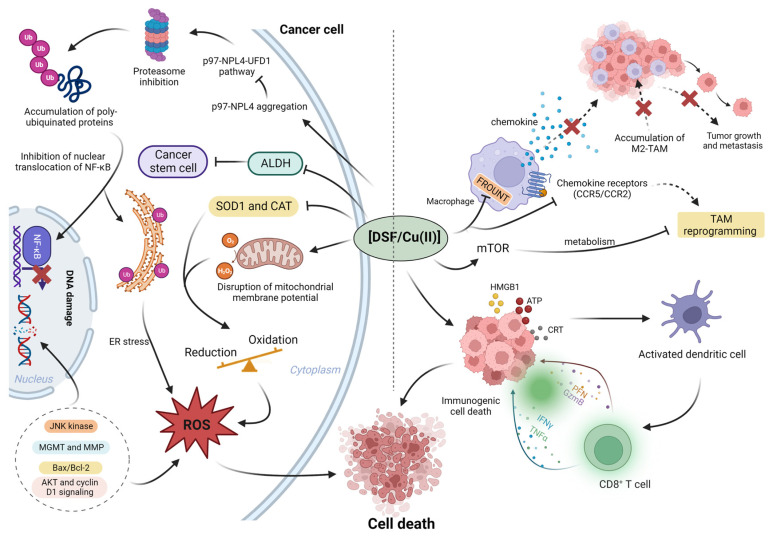
The summary of roles of DSF/Cu (II) in cancer microenvironment. Left: DSF/Cu (II) inhibits the cancer proteasome activity via p97-NPL4 pathway; in addition, DSF/Cu (II) inhibits cancer-associated ALDH activity and inhibits cancer stem cells (CSCs). In the cancer microenvironment, aberrant enzyme activity, superoxide dismutase 1 (SOD1) and catalase (CAT), results in the elevation of ROS; the higher basal level of ROS benefits cancer proliferation. However, the further increased ROS to exceed cancer tolerance cause cancer death. Right, the DSF/Cu (II) reprograms the tumor-promoting macrophage M2 to anti-tumor type M1. In addition, DSF/Cu (II) transforms the immune-suppressive (cold) tumor microenvironment to the immune-active (hot) microenvironment via the induction of immunogenic cell death (ICD). Created with BioRender.com. accessed on 19 April 2023.

**Figure 4 pharmaceutics-15-01567-f004:**
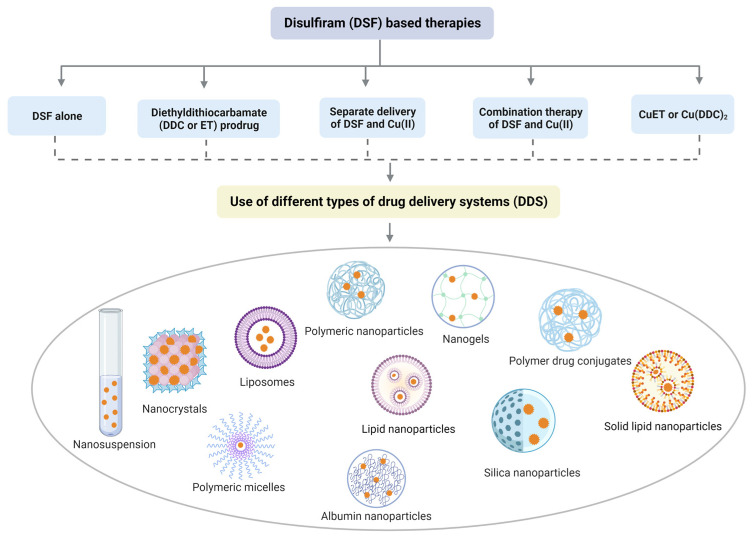
Divergent nanotechnology and chemical modulation-based formulations to enhance DSF anticancer effects. To sum up, DSF-based nanomedicine includes DSF alone, DDC prodrug delivery system, delivery system for Cu (II) and DSF/Cu (II), and drug delivery system for active component-CuET.

**Figure 5 pharmaceutics-15-01567-f005:**
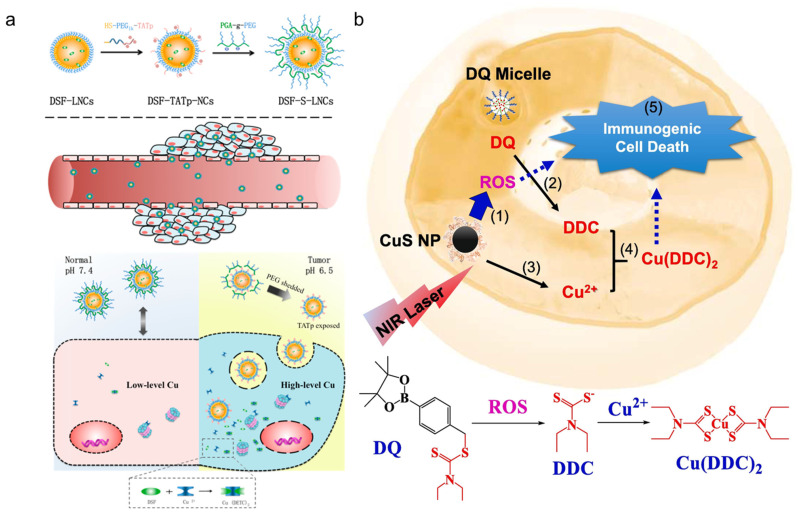
Typical examples of the DSF delivery system and DDC prodrug: (**a**) DSF-loaded pH-triggered PEG-shedding TAT peptide-modified lipid nano capsules [160]. Copyright 2015, American Chemical Society. (**b**) Near-infrared light triggered activation of pro-drug combination cancer therapy and induction of immunogenic cell death, (1) NIR laser + CuS NP treatment increases intracellular ROS. (2) ROS converts DQ prodrug to DDC. (3) CuS NP release Cu (II). (4) DDC and Cu (II) form Cu(DDC)_2_ active anticancer complex. (5) Cu(DDC)_2_ chemotherapy and ROS induce immunogenic cell death in cancer cells [101]. Copyright 2015 © 2021 Elsevier B.V.

**Figure 6 pharmaceutics-15-01567-f006:**
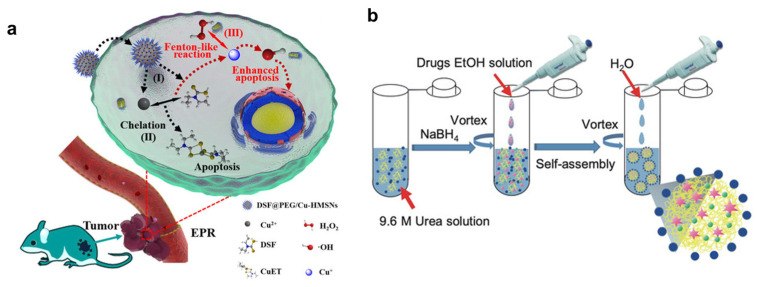
(**a**) Schematic illustration: The DSF@PEG/Cu-HMSNs nano system triggers tumor-specific DTC-Cu (II)chelation (pathway I to II) and Cu+-initiated Fenton-like reaction (pathway I to III). This enhances the antitumor efficacy of DSF-based chemotherapy without systemic toxicity. Copyright 2019, American Chemical Society [174]. (**b**) Schematic illustration of albumin-based dual-targeting biomimetic delivery of Rego and DSF/Cu for cancer therapy. Copyright 2021, Wiley-VCH [192].

**Figure 7 pharmaceutics-15-01567-f007:**
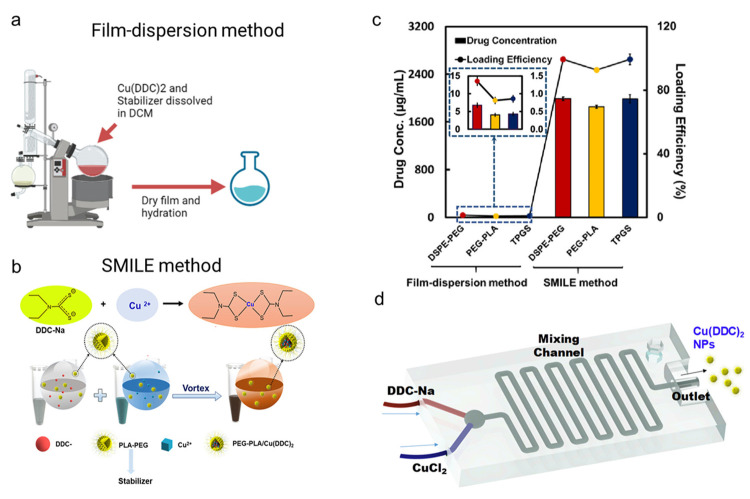
Schematic comparison of (**a**) film-dispersion method (Created with BioRender.com) accessed on 8 May 2023 vs. (**b**) stabilized metal ion ligand complex (SMILE) technology [25]. Copyright © 2018 American Chemical Society. (**c**) Drug loading efficiency of film-dispersion method and stabilized metal ion ligand complex (SMILE) technology [25]. Copyright © 2018 American Chemical Society. (**d**) The scale of SMILE technology using a 3D-printed microfluidic device [32]. Copyright © 2019 Elsevier Ltd.

**Figure 8 pharmaceutics-15-01567-f008:**
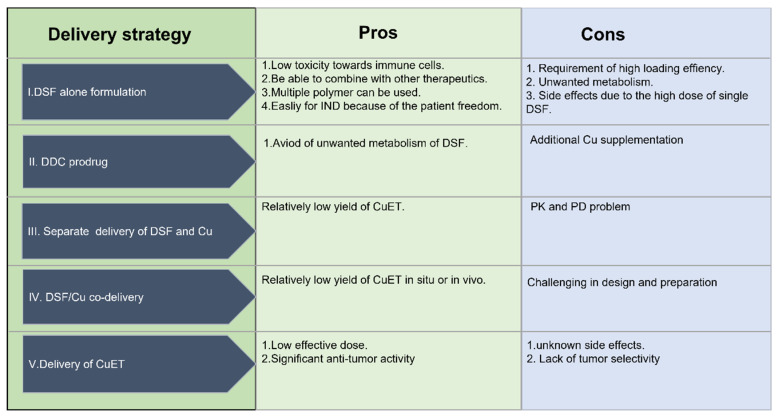
The pros and cons of the different delivery strategies.

**Table 1 pharmaceutics-15-01567-t001:** A summary of studies on DSF-based clinical trials (http://clinicaltrials.gov) accessed on 10 April 2023.

Cancer	Status	Clinical Identifier (Clinicaltrials.Gov)
Metastatic melanoma	Phase I, Terminated	NCT00571116
Melanoma	Phase I/II, Completed	NCT00256230
Melanoma	Phase II, Completed	NCT02101008
Prostate Cancer	Phase I, Completed	NCT01118741
Prostate Cancer	Phase I, Recruiting	NCT02963051
Breast Cancer (Metastatic)	Phase II, Recruiting	NCT03323346
Refractory Breast Cancer (Metastatic)	Phase II, Recruiting	NCT04265274
Pancreatic Cancer (Metastatic, Recurrent)	Phase I, Recruiting	NCT02671890
Pancreatic Cancer (Metastatic)	Phase II, Not Yet Recruiting	NCT03714555
Recurrent Glioblastoma	Phase I, Active, Not Recruiting	NCT02770378
Glioma Glioblastoma	Phase II/III, Recruiting	NCT02678975
Glioblastoma Multiforme	Phase II, Recruiting	NCT03363659
Solid Tumors Involving Liver	Phase I, Completed	NCT00742911
Non-small Cell Lung Cancer	Phase II/III, Completed	NCT00312819
Glioblastoma (Recurrent)	Phase II, Completed	NCT03034135
Glioblastoma	Phase I/II, Recruiting	NCT02715609
Glioblastoma	Phase II, Not Yet Recruiting	NCT01777919
Glioblastoma	Phase II/III, Recruiting	NCT02678975
Glioblastoma	Early Phase I, Recruiting	NCT03151772
Glioblastoma	Early Phase I, Completed	NCT01907165
Germ Cell Tumor	Phase II, Recruiting	NCT03950830
Multiple Myeloma	Phase I, Terminated	NCT04521335
Refractory Sarcomas	Phase I, Recruiting	NCT05210374
Advanced Gastric Cancer	Phase Not Defined, Not Yet Recruiting	NCT05667415

**Table 2 pharmaceutics-15-01567-t002:** Drug delivery systems of combined DSF/Cu, DSF alone, DDC prodrug, and Cu alone.

Drug	Nanoparticle Delivery System	Cancer Type/Cell Lines	Important Findings	Reference
**DDS for DSF alone**	Disulfiram-loaded biotin-mediated PEGylated nanostructured lipid	Breast cancer (4T1 cells)	These nanoparticles, when coupled with copper ions, shown enhanced accumulation in tumors and efficiently inhibited tumor growth in breast xenograft mice model.	[151]
	DSF-loaded vitamin E-TPGS-modified PEGylated nanostructured lipid carriers	Breast cancer (4T1 cells)	These nanoparticles showed significantly higher tumor growth inhibition rates of (48.24%) compared to free DSF (8.49%) and DSF-NLC (29.2%) formulations.	[149]
	DSF-loaded redox-sensitive shell crosslinked micelles	Breast cancer (4T1 cells)	These nanoparticles demonstrated a remarkable ability to inhibit tumor growth and prevention of lung metastasis of 4T1 tumors.	[117]
	Disulfiram-loaded soy-protein-isolated nanosuspension	Breast cancer (MDA-MB-231 cells)	These nanosized, sphere shaped NPs exhibited higher drug loading capacity, increased entrapment efficiency, improved stability, sustained release, higher in vitro cellular uptake, and were found to be more cytotoxic compared to free solution of DSF.	[152]
	Disulfiram- and doxorubicin-loaded polycaprolactone-b-poly(L-glutamic acid)-g-methoxy poly(ethylene glycol) nanoparticles	Breast cancer (MCF-7 and MDA-MB-237 cells)	These NPs efficiently accumulated in tumors, indicating their effective targeting ability.Moreover, when compared to the free DSF, the NPs showed improved synergistic effect on antitumor activity.	[153]
	mPEG-PLGA/PCL mixed nanoparticles	Breast cancer (4T1 cells)	NPs showed high stability in both water and 10% serum-containing PBS, which indicate the integrity under physiological status. In addition, the nps enhanced disulfiram levels in the blood, and efficiently inhibit the growth of 4T1 murine xenograft tumors.	[154]
	Folate receptor targeted PLGA-PEG nanoparticles	Breast cancer (MCF-7 cells)	These nanoparticles showed higher apoptosis induction compared with free drug. Moreover, the NPs showed dose-dependent inhibition of caspase-3 but produce concentration-independent cell cycle arrest at G0/G1 and S-phase.	[155]
	Disulfiram-loaded PLGA-PEG nanoparticles	Breast cancer (MCF-7 cells)	Folate receptor targeted nanoparticles induced ROS formation, which benefits the cancer apoptosis. When compared to untargeted nanoparticles.Modified nanoparticles decreased cell proliferation and tumor growth rate more efficiently	[156]
	Disulfiram-encapsulated PLGA nanoparticles	Breast cancer (MCF-7 cells)	This delivery system prevented rapid degradation of DSF and provide sustained release in tumor cells. Moreover, these NPs induced apoptosis more efficiently compared to free disulfiram.	[157]
	DSF-encapsulated PLGA nanoparticles	Liver cancer (PLC/PRF/5 and Huh7 HCC cells)	These nanoparticles significantly inhibited liver cancer stem cell population and demonstrated anti-metastatic effect in liver cancer xenograft mouse model.	[158]
	Disulfiram-loaded polysorbate 80-stabilized PLGA nanoparticles	Liver cancer (Hep3B cells)	These nanoparticles inhibited cell proliferation via cell cycle arrest and activation of apoptotic pathways In addition, the PLGA np ensure sustained the drug release, thereby potentially lowering the dosage regimens.	[159]
	DSF-loaded PEG-shedding lipid nanocapsules	Liver cancer (Hep G2 cells)	These nanoparticles showed 74.5% higher delivery efficiency compared with lipid nanocapsules alone in liver cancer xenograft-bearing mice model.	[160]
	Disulfiram encapsulated mixed (mPEG_5000_-PCL_5000_) nanoparticles	Liver cancer (H22 cells)	These nanoparticles significantly inhibited tumor growth rate and showed greater magnitude of tumor cell necrosis compared with DSF solution.	[161]
	Hybrid paclitaxel–DSF nanocrystals	Lung cancer (A549 cells)	These hybrid nanoparticles showed 6-fold increase in apoptosis and 12-fold decrease in tumor volume in resistant lung cancer xenograft mice model.	[162]
	Disulfiram-loaded PLGA nanoparticles	Lung cancer (A549 cells)	The evaluation indicatedthat increasing the amount of drug input to carrier, molecular weight of stabilizer, as well as the sonication time reduced the size of nanoparticle. Moreover, the np protect the DSF from clearance, thereby increasing the disulfiram cytotoxicity.	[163]
	Aminated mesoporous silica nanoparticles	Lung cancer (A549 human non-small cell lung carcinoma cells)	These nanoparticles showed excellent cytotoxicity profiles, exhibited substantial suppression of tumor volume, and compared to free DSF, the NPs shown. limited adverse effects.	[164]
	DSF-loaded biodegradable monomethoxy (polyethylene glycol) d, l-lactic co-glycolic acid (mPEG-PLGA) nanoparticles	Brain cancer (DAOY and T98G human brain cancer cells)	These nanoparticles showed favorable inhibition of intracranial medulloblastoma xenografts compared to unencapsulated DSF.	[165]
	Disulfiram- and folic-acid-incorporated metal organic framework (IRMOF3-DSF-FA)	Oral cancer (Cal27 and HACAT cells)	These nanoparticles showed favorable biocompatibility and greater cellular uptake, targeted tumor tissues, and effectively inhibited ALDH1+ cancer stem cells with no damage to vital organ.	[166]
**DDS for DDC prodrug**	DSF prodrug and copper sulfide nanoparticles+ near Infrared laser combination therapy	Breast cancer (4T1 cells)	This combination therapy effectively increased the intra-tumor ROS levels, which efficiently activated DQ prodrug. This combination also induced immunogenic cell death, thereby being a inducer for eliciting antitumor immunity.	[101]
	H_2_O_2_-responsive diethyldithiocarbamate-based prodrug	Breast cancer (4T1 cells)	DQ showed much lower cytotoxicity (IC_50_ > 100 µM) to normal cells than DSF (IC_50_ of 12.5 µM), suggesting the advantage of DQ.	[167]
	Dithiocarbamate releasing prochelator GGT-DTC, which requires activation by γ-glutamyl transferase (GGT)	Prostate cancer (22Rv1, LNCaP, PC3 prostate cancer cells, as well as PWR-1E prostate epithelial cells)	GGT-DTC shown favorable stability against non-specific degradation in both normal and prostate cancer cells. and GGT-DTC selectively released diethyldithiocarbamate only in cells with measurable GGT activity.	[168]
	Β-D-galactose receptor targeted disulfiram-loaded nanoparticles	Ovarian cancer (SKOV-3 ovarian cancer cells and NCI-Adr-Res drug-resistant ovarian cancer cells)	These nanoparticles, upon internalization by cells, degrade and release diethyldithiocarbamate due to cleavage of disulfide bonds and form Copper (II)DDTC complex, which showed much greater tumor mass penetrating and destructive capacity. In addition, these NPs exhibited greater tumor growth inhibition capacity than the dosage form used in clinical trials (DSF in combination with copper gluconate).	[169]
**DDS for Copper**	N-Oxide polymer-cupric ion nanogels	Breast cancer (MDA-MB-231 and 4T1 cancer cells)	These neutral and water soluble zwitterionic N-oxide polymer, poly [2-(N-oxide-N,N-dimethylamino)ethyl methacrylate/Cu nanogels efficiently delivered copper ions to tumor cells both in vitro and in vivo levels. The effective delivery of copper potentiated antitumor activity of DSF.	[150]
	Ferritin-albumin-Cu nanoparticle in combination with disulfiram	Breast cancer (4T1, MDA-MB-231 cells)	These nanoparticles shown favorable accumulation in the tumor and demonstrated targeting capacity towards cancer cells In vivo assays, the NPs also shown more potent anti-tumor efficacy compared to DSF or nanoparticle alone.	[170]
	Copper-cysteamine nanoparticles	Esophageal cancer (Human ESCC KYSE-30 cells)	These nanoparticles showed greater inhibition of tumor growth compared to DSF and Cu-Cy alone, and resulted in ROS accumulation, and blocked nuclear translocation of NF-kB in esophageal cancer cells.	[171]
**DDS for combined DSF/Cu**	Glutathione-responsive coordination nanoparticles (Cu-IXZ@DSF)	Breast cancer (4T1 cells)	These nanoparticles showed good biosafety and excellent antitumor activity via the increase of endoplasmic reticulum (ER) stress.	[172]
	BSA/Cu(DDC)_2_ metal organic nanoparticles	Breast cancer (4T1 cells)	This study shown the scale-production using 3D printing device, the NPs generated by the device showed potent antitumor activity and effectively inhibited growth of tumors in orthotopic 4T1 breast cancer mice model.	[32]
	Fe3O4@mSiO2 magnetic mesoporous silica nanoparticles	Breast cancer (MCF-7 cells)	The cytotoxicity of these DSF-loaded carrier systems was improved by adding copper and/or sodium nitroprusside, and cytotoxicity of NPs was greater in MCF-7 cells compared to non-tumorigenic MCF-10A cells.	[173]
	Disulfiram in combination with bacterially synthesized copper oxide nanoparticles	Breast cancer (MDA-MB-231 cells)	These combination nanoparticles showed higher pro-oxidant effect-mediated apoptosis and anti-metastatic potential via inhibition of antioxidant defenders and elevation of cellular reactive oxygen species.	[157]
	Copper-doped DSF-loaded hollow mesoporous silica nanoparticles	Breast cancer (4T1 cells)	These nanoparticles showed high chemotherapeutic efficacy with tumor growth inhibition (TGI) values as high as 71.4% compared to free DSF (which did not show antitumor effect).	[174]
	pH-responsive metal organic framework nanoparticles (DSF/DOX@ZIF-8@Cu-TA)	Breast cancer (MDA-MB-231 cells)	These nanoparticles significantly enhanced therapeutic efficiency of DSF and DOX both in vitro and in vivo. Accumulation of DSF and Cu (II) resulted in rapid formation of highly cytotoxic complexes accompanied with the generation of ROS.	[175]
	Mannosylated albumin nanoparticles with co-encapsulation of DSF/Cu and regorafenib	Colorectal cancer (Human colon cancer drug-resistant HCT8/ADR cell line, mouse fibroblast L929 cells)	This combination therapy greatly suppressed the growth of drug-resistant colon tumors, enhanced apoptosis, and upregulated intracellular ROS and anti-angiogenesis.	[176]
	Disulfiram-loaded hollow copper sulfide nanoparticles	Colorectal cancer (CT26 cells)	These nanoparticles +NIR laser significantly induced apoptosis with72% in vitro and 100% in vivo. Furthermore, the treatment approach effectively promoted tumor elimination in vivo.	[177]
	Disulfiram- and copper-loaded pH-responsive lipid-coated calcium phosphate nanoparticles	Colon cancer (CT26 murine colon cancer cells)	These nanoparticles effectively induced the immunogenic cell death of cancer cells, thereby contributing to enhancement immune checkpoint blockade therapy.	[178]
	Copper-ion- and disulfiram-loaded hyaluronic acid (HA)/polyethyleneimine (PEI) nanoparticles	Esophageal cancer (Eca109 and TE1 Esophageal squamous cell carcinoma cells)	These nanoparticles showed higher apoptosis than 5-FU (a conventional therapeutics), DSF/Cu and control and inhibited tumor proliferation with no toxicity on normal tissues.	[179]
	Combination of Disulfiram with copper oleate PEGylated liposome	Liver cancer (H-22 cells)	These nanoparticles demonstrated prolonged circulation, increased area under curve (AUC) and an increase in tumor inhibition rates by producing synergistic antitumor effect.	[180]
	Hyaluronic-acid-decorated liposomes containing Cu(DDC)_2_	Pancreas cancer (Human Panc1 pancreatic adenocarcinoma cells)	These nanoparticles produced high ROS-mediated anticancer efficacy and increased anti-proliferative activity on pancreatic cancer stem cells, compared to DSF, Zn(DDC)_2_ and Fe(DDC)_2_.	[181]
**DDS for CuET**	Copper–drug complexes in liposomes	Breast cancer (MDA231-BR cells)	Synthesis pf Cu(DDC)_2_ in lipid vesicles enhanced the stability and addressed the solubility issues related to each agent.	[182]
	CuET/DIR (near infrared dye) nanoparticles	Breast cancer (MCF-7, 4T1, and 4T1 subline-LG12 cells)	These nanoparticles showed enhanced tumor killing efficacy through nuclear targeting compared to CuET alone and showed optimal biocompatibility.	[183]
	Copper(II)-disulfiram-loaded melanin dots	Breast cancer (4T1 and mouse fibroblast NIH/3T3 cells)	These nanodots showed good tumor accumulation, excellent tumor inhibition capacity, and higher tumor growth inhibition rate of 45.1%; The combination with photothermal the produce a higher tumor growth inhibition rate (78.6%) compared to nanodots without irradiation.	[184]
	DTC–copper complex@hyaluronic acid nanoparticles	Breast cancer (M231 cells)	These nanoparticles produced significant cytotoxicity towards cancer cells, The nanoparticles accumulated in tumors, and elicited tumor growth inhibition at a dose of 1mg/kg without toxic side effects.	[185]
	Stabilized metal ion ligand complex (SMILE) to prepare Cu(DDC)2 nanoparticles	Prostate cancer (DU145-TXR cells)	These nanoparticles induced cell death in drug-resistant prostate cancer cell lines through paraptosis (Paraptosis is a type of programmed cell death, distinct from apoptosis or necrosis, paraptosis involves the dilation of mitochondria, formation of vacuoles in the cytoplasm, and swelling of organelles, leading to cell death.).	[25]
	Injectable copper diethyldithiocarbamate formulation	Brain cancer (F98 glioblastoma cells)	This formulation showed ~50% reduction in tumor volume at its respective maximum tolerated dose compared to vehicle- and copper-treated animals.	[186]

## Data Availability

Not applicable.

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
