# Peer review of "Advancing Cancer Therapy with Copper/Disulfiram Nanomedicines and Drug Delivery Systems"

_pharmaceutics, 2023, doi:10.3390/pharmaceutics15061567_

Round 1

Reviewer 1 Report

The review article entitled“ Advancing Cancer Therapy with Copper/Disulfiram Nanomedicines and Drug Delivery Systems” is focused on advances in the anticancer effect of Copper/Disulfiram based drug delivery and Nanomedicines. The writing and summary of literature articles were well represented. The article was designed and organized. The result and discussion are very well-written findings of the reported literature. The Intro, results, and discussion sections are very well written. The figures and tables included are adequate, and the author also draws a few figures to explain the concept and mechanism.

Overall, I recommend for Accept this MS after minor revisions. This is a high-quality article and will be significant literature in the field.

My specific comments for minor review are:

Under 4. DSF-based therapies for cancer treatment, summarizing the DSF-based therapies findings in various cancer types, will further enhance MS's MS significance.

Author Response

Attached is the responses to the comments of the reviewer.

Reviewer 2 Report

The authors reviewed the recent progress of copper/disulfiram nanomedicine for cancer therapy. Among which, the mechanism of copper and disulfiram based anti-tumor strategy was summarized clearly, and the representative nanomedicine for the delivery of copper and disulfiram was also analyzed. Overall, the manuscript was well organized and easy understanding. To improve the manuscript, the schematic figures of the representative works should be added, and the formation of the nanomedicine was suggested to be added in the schematic figures for better understanding.

The quality of english language was acceptable for reading.

Author Response

(The authors gave the same response as above.)

Reviewer 3 Report

The manuscript by Chen and co-workers is a concise overview of the thriving research field of drug delivery systems, focusing of the use of copper/disulfiram nanomedicines for cancer treatment. 

I believe the article should be published practically as it is. However, it would be great if the authors take into consideration the following comments/additions:

1.     In the abstract, include what the abbreviations CSCs and DDC refer to.

2.     In line 32, delete "." after the sentence "drug resistance".

3. Which metabolite causes the transformation of DSF into the primary metabolite (DDC), that is who breaks a disulphide bond. Please, add to Figure 1. 

4. A brief discussion (perhaps in the conclusion section) on some of the present and future challenges of the field and how such feature could affect (pre)clinical applications, and the importance of knowing the different courses of action before, during and after their applications.

Author Response

(The authors gave the same response as above.)
